# Relationship of Neuropeptide S (NPS) with Neurocognitive, Clinical, and Electrophysiological Parameters of Patients during Structured Rehabilitation Therapy for Schizophrenia

**DOI:** 10.3390/jcm11185266

**Published:** 2022-09-06

**Authors:** Agnieszka Markiewicz-Gospodarek, Renata Markiewicz, Beata Dobrowolska, Mansur Rahnama, Bartosz Łoza

**Affiliations:** 1Department of Human Anatomy, Medical University of Lublin, 20-090 Lublin, Poland; 2Department of Neurology, Neurological and Psychiatric Nursing, Medical University of Lublin, 20-093 Lublin, Poland; 3Department of Holistic Care and Management in Nursing, Medical University of Lublin, 20-081 Lublin, Poland; 4Department of Oral Surgery, Medical University of Lublin, 20-093 Lublin, Poland; 5Department of Psychiatry, Medical University of Warsaw, 02-091 Warsaw, Poland

**Keywords:** neuropeptide S, schizophrenia, cognitive functioning, rehabilitation, AEP, QEEG

## Abstract

Introduction: Neuropeptide S is a biomarker related to various neuropsychiatric and neurocognitive functions. Since the need to improve cognitive functions in schizophrenia is unquestionable, it was valuable to investigate the possible relationships of plasma levels of NPS with neurocognitive, psychopathological and EEG parameters in patients with schizophrenia. Aim: Relationships between the serum NPS level and neurocognitive, clinical, and electrophysiological parameters were investigated in patients diagnosed with schizophrenia who underwent structured rehabilitation therapy. Methods: Thirty-three men diagnosed with schizophrenia were randomized into two groups. The REH group (N16) consisted of patients who underwent structured rehabilitation therapy, the CON group (N17) continued its previous treatment. Additionally, the reference NPS serum results were checked in a group of healthy people (N15). In the study several tests assessing various neurocognitive functions were used: d2 Sustained-Attention Test (d2), Color Trails Test (CTT), Beck Cognitive Insight Scale (BCIS), Acceptance of Illness Scale (AIS), and General Self-Efficacy Scale (GSES). The clinical parameters were measured with Positive and Negative Syndrome Scale (PANSS) and electrophysiological parameters were analyzed with auditory evoked potentials (AEPs) and quantitative electroencephalography (QEEG). The NPS, neurocognitive, clinical, and electrophysiological results of REH and CON groups were recorded at the beginning (T1) and after a period of 3 months (T2). Results: A decreased level of NPS was associated with the improvement in specific complex indices of d2 and BCIS neurocognitive tests, as well as the improvement in the clinical state (PANSS). No correlation was observed between the level of NPS and the results of AEPs and QEEG measurements. Conclusions: A decreased level of NPS is possibly related to the improvement in metacognition and social cognition domains, as well as to clinical improvement during the rehabilitation therapy of patients with schizophrenia.

## 1. Introduction

While Bleuler assumed from the very beginning that schizophrenia has a multilayered and especially neurocognitive origin, his idea was updated at the end of 20th century by Andreasen et al., in the neo-Bleulerian cognitive dysmetria model and served as an umbrella concept for the dozens of more specific cognitive hypotheses for the origins of schizophrenia, such as neurodevelopmental theory, frontal dysconnectivity, dorsolateral hypo-frontality, sensory gating deficits, and other domain-specific models [1,2,3]. All such “misconnection” theories identify the disturbances in basic cognitive and behavioral domains (e.g., speed of processing, attention, working memory, learning, and problem solving), but at the same time their biological substrates remain at best in the realm of well-formulated but isolated observations [4]. As a result, there are currently no neurocognitive methods enabling individual and specific schizophrenia diagnosis, and no approved pro-cognitive drugs or therapies exist [5].

Various abnormalities of the neuropeptide system have been demonstrated in schizophrenic patients [6]. However, there is no evidence of a true primary or secondary relationship between neuropeptides and schizophrenia that contributes directly to its etiopathogenesis. On the other hand, the need to improve cognitive functions in schizophrenia is fundamental and unquestionable and there are studies showing the beneficial effect of neuropeptides on cognitive functioning. Neuropeptides and typical neurotransmitters are usually co-released; however, the neuropeptides require high-frequency burst firing, which enables both the coordinated and independent neurotransmitter activity [6]. Neuropeptides act through metabotropic G-protein-coupled receptors (GPCRs), changing cell excitability, signaling, and gene expression. 

Neuropeptide S (NPS) has multiple neuropsychiatric functions [7], and it has been postulated that it may play an important role in regulating cognitive functioning [8]. The problem is that so far, the studies have been conducted almost exclusively on animals [9]. Neuropeptide S (NPS) is a 20-amino-acid ligand, which name originates from its N-terminal serine residue. It is found in human beings and nearly all tetrapods [10,11]. The precursor mRNA of NPS is found in only a limited number of regions of the brain (trigeminal nucleus, lateral parabrachial nucleus, locus coeruleus, and amygdala), and in contrast, NPSR1 mRNA is widely expressed in the entire central nervous system (CNS) [11,12]. NPS fibers project to limbic and thalamic areas such as the amygdala, hypothalamus, and paraventricular thalamic nucleus [13]. In humans, the neurons expressing NPS and NPSR1 mRNA were mainly found in the regions important for integration of autonomous information and emotional behavior like the parabrachial area [14]. 

Preclinical and clinical studies of the NPS/NPSR1 system have remained separated thus far, and there is no comprehensive description of the role of this system neither in humans nor in rodents [11]. The NPS/NPSR1 system seems to play a significant role in stress responsiveness and the activation of the hypothalamic–pituitary–adrenal axis in rodents [11,15]. NPS activity is associated with inhibitors of neurons which gate the amygdala output [16]. The NPS/NPSR1 system also participates in regulating the wakefulness–sleep cycle [17]. It is, therefore, assumed that since the NPS metabolism is highly conservative across different species, research on animals may be accurately extrapolated to humans [16]. While such assumptions can be true in the case of the behavioral regulation of anxiety [10], arousal [10], or pain [16,18], it is difficult to simply extrapolate this way with the assumed role of NPS/NPSR1 in drug addiction [19,20], memory consolidation, conceptual generalization [19], or especially personality formation [20]. 

The therapeutic use of the NPS/NPSR1 system in humans has been suggested since the discovery of NPS [10]. NPS/NPSR1 activity could potentially be useful in the treatment of various anxiety disorders [21]. The authors of animal studies predicted that the NPS/NPSR1 system would facilitate the extinction of conditioned fear [11,22]. Specifically, the anxiolytic effect is not related to excessive sedation, but rather to an increase in activity (“novel activating anxiolytic”), which is a pharmacologically unique feature [21]. The median plasma NPS level was found to be significantly higher in generalized anxiety disorder (GAD) patients [23]. While NPS may have a beneficial effect on anxiety, no direct effect on depression has been demonstrated thus far in animal models [24]. The NPS/NPSR1 system could be the target for the development of drugs for wakefulness–sleep disorders [16], to alleviate motor and non-motor dysfunctions of Parkinson’s diseases [12], to improve learning and memory, e.g., in Alzheimer’s disease [9], and to treat substance abuse disorders [25,26]. 

There are only preliminary data on the relationship between the NPS/NPSR1 system and the course of schizophrenia [7]. A case–control comparison revealed that the low functioning NPSR1 Asn107 variant was significantly associated with schizophrenia [27]. However, another study revealed no genetic association of NPSR1 alleles with schizophrenia (and ADHD), suggesting a rather specific relationship of NPSR1 with anxiety disorders [28]. There are various separate animal patterns for specific dysfunctions that could support the diagnostic and/or therapeutic potential of the NPS/NPSR1 system in schizophrenia research, for example, the acoustic startle response [27], but there is no comprehensive animal model to directly transfer these data to human pre-clinical or clinical models. The mechanism of the psychopharmacological effect of NPS on schizophrenia psychopathology may result from blocking the NMDA antagonist-induced deficits in prepulse inhibition [27,28,29]. NPS blocks MK-810 NMDA antagonism, suggesting a potential antipsychotic effect of NPS, such as MK-801, which blocks NMDA transmission and serves as a pharmacological model of schizophrenia [28,29]. Nevertheless, the similarity of NPS to anti-psychotics is not complete as haloperidol and sulpiride, both being dopamine D2 receptor antagonists, inhibit NPS-induced anti-nociceptive activity [17]. Long-term olanzapine administration led to the upregulation of NPS and downregulation of the NPSR expression in the rat hypothalamus [30]. Chronic haloperidol administration led to the upregulation of NPS and NPSR in the rat brainstem [31]. These animal results suggest that anti-psychotics may work by affecting peptidergic signaling. However, they do not provide answers about the real impact of the NPS/NPSR1 system on schizophrenia.

The impact of intensive rehabilitation, especially with the use of the neurofeedback (NF) technique, on the level of peptide factors such as BDNF, as well as on the clinical state, has already been shown in human studies [32,33]. However, no studies on the relationship between plasma NPS in patients with schizophrenia and any type of treatment have been published so far. Although investigations of NPS’s permeability from the blood–brain barrier have not been conducted on human subjects, the rationality of measuring plasma NPS level in patients with mental disorders has been demonstrated [23]. 

The purpose of the study was to examine the relationship between NPS serum level cognitive parameters during the structured rehabilitation therapy of patients with schizophrenia. In addition, this evaluation was performed in relation to the clinical condition and results of electrophysiological tests. 

## 2. Materials and Methods

### 2.1. Study Design 

This study was a randomized, controlled 3-month trial reported with the use of CONsolidated Standards of Reporting Trials (CONSORT) guidelines [34]. The trial is registered in the ISRCTN registry (Trial ID: ISRCTN78612833) where the full protocol can be found. Thirty-three male patients with paranoid schizophrenia (according to ICD-10-DCR [ICD]) were divided into two groups: a group in an intensive rehabilitation programme (REH, N16), and a control group with standard social support (CON, N17). In the study, the sample size (N) was calculated for the test power of the NPS level in the range of not less than 0.8, which criterion is considered strong and adequate in behavioral sciences [35]. 

The N for 0.8 test power was set to 15 and respectively: 16 for 0.83 and 17 for 0.85. Members of both REH and CON groups were recruited from participants of a city day-care center programme. They continued their antipsychotic treatment and usual clinical management. Additionally, a group of healthy (H), non-clinical males (N15) with comparable characteristics was considered to check NPS reference results.

### 2.2. Participants

The inclusion criteria (CON and REH groups) were patients’ consent, male gender, clinical diagnosis of paranoid schizophrenia [ICD], age 18–50, right-handedness (writing), no current neurological diseases, mental disability, or alcohol and/or psychoactive sub-stance addiction. The inclusion criteria in the non-clinical group (H) were the same as above, but all the participants were mentally healthy. The study was limited only to male participants to reduce the risk of potential gender differences in NPS levels which could not be corrected reliably between relatively small groups. Previous NPS studies with a limited number of participants clearly indicated difficulties in interpreting the results in relation to gender [23,24,28]. Moreover, PANSS results can also be influenced by gender differences [36]. 

Subjects, after meeting the inclusion criteria, were randomly assigned to two groups (REH, CON), without the researchers participating in the drawing process. 

All recruited patients had remained relatively stable, i.e., without active psychotic episodes for not less than 18 months. The patients cannot be treated as clinically “residual” according to ICD-10-DCR, as they were quite young, active, and multi-episodic, so they fit the pattern of episodic schizophrenia with stable or progressive development of negative symptoms in the intervals between psychotic episodes (ICD-10-DCR: F20.01/F20.02) [8]. No current suicidal risk was diagnosed. 

As can be seen from Table 1A, all the significant study parameters were not statistically different at the baseline: PANSS Total, PANSS Positive, PANSS Negative, PANSS General, age at the first hospitalization, NPS serum level, BMI (body mass index), and age of participants. Group comparisons were presented in Table 1B. The H statistic was not significant in terms of any parameter in Table 1B, indicating that all groups were from the same distribution.

Patients from the CON group had on average three previous psychiatric hospitalizations (M 2.77, SD 1.60), and the REH group-four (M 4.19, SD 1.17). Almost all the patients lived on a disability pension or other social benefits. A significant proportion of the participants smoked cigarettes: CON—76.5%, REH—56.3%, and Non-clinical—66.7%.

During the experiments, all patients continued their former antipsychotic treatment (daily dose olanzapine equivalents in milligrams: CON vs. REH: M 19.32 SD 4.97 vs. M 21.28 SD 6.88). The antipsychotic treatment pattern was not changed during experiment. All subjects were administered atypical antipsychotics (olanzapine, clozapine, quetiapine, risperidone, aripiprazole), and only some of them additionally received typicals (sulpiride, perazine, zuclopenthixol, flupenthixol, haloperidol). On average half of the study participants were subjected to monotherapy (only with atypical antipsychotics): REH group—9/16, CON group—8/17. Polytherapy was delivered with either 2 or more atypical antipsychotics (REH 4/16, CON 7/17) or a combination of atypical and typical antipsychotics (REH 3/16, CON 2/17). Chi-squared test for those three observations (atypical monotherapy, atypical polytherapy, and atypical/typical polytherapy) between REH and CON was insignificant (χ2 = 1.91, df = 2, *p* = 0.385). None of the patients had taken anticholinergic drugs.

### 2.3. Outcome Measures 

The examinations were performed twice, at the beginning (T1) and after a period of 3 months (T2).

#### 2.3.1. Neurocognitive Tests 

##### d2 Sustained-Attention Test (d2)

The d2 test was used to measure a patients’ cognitive performance, including attention, concentration endurance, execution speed, and ability to correct errors [37]. The test consists of 14 lines with 47 characters in each line. Participants have 20 s per line to cross out all lower-case d’s with two apostrophes above or below the letter. Every 20 s, the subject moves on to the next line. There are various descriptive and complex indices of d2 results [38,39]:
TN—the total number of letters marked both correctly and incorrectly; the speed of processing score; E—raw score of omission and commission errors; the attention carelessness and confusion score; E%—percentage of all errors; the overall accuracy score; TN-E—the total number of items processed minus all errors; the impact of attention on the combined scores of speed and accuracy as a perception ability; CP—the concentration performance, the number of correctly processed items minus the commission errors; FR—the fluctuation rate which is based on the difference in correct responses between the rows with the highest and lowest number of correct responses. 

##### Color Trials Test (CTT) 

The CTT is comprised of two different tasks [40]. First, the respondent must connect circles in an ascending numbered sequence (1–25; CTT-1). Then, the task is to connect numbers in an ascending sequence (1–25) while alternating between pink and yellow colors, ignoring the distracter color (CTT-2). CTT-1 and CTT-2 were developed to measure sustained and selective types of attention, visual spatial skills, and motor speed. CTT-2 is also dedicated to the cognitive assessment of Stroop-like effects based on mental flexibility–constantly reloaded tasks in the executive memory [41]. 

The Interference Index (CTT-II) is a difference between CTT-2 and CTT-1 time divided by CTT-1, what provides information about the increase in the relative time needed to perform a task with a higher degree of cognitive complexity.

##### Beck Cognitive Insight Scale (BCIS) 

The BCIS is a complex 15-item self-report designed to estimate two aspects of cognitive insight in psychotic patients: the Self-Reflectiveness (9 items; BCIS-REF) and the Self-Certainty (6 items; BCIS-CER) [42]. By subtracting the Self-Certainty from the Self-Reflectiveness, the composite Reflectiveness–Certainty Index (BCIS-INDEX) score can be obtained, which is a balanced measure of cognitive insight.

##### Acceptance of Illness Scale (AIS) 

The AIS consist of eight statements, each graded from 1 to 5 [43]. Its higher score is indicative of better disease acceptance. The AIS examines not only whether the patient “knows” that he or she has schizophrenia, but mostly the perception of a disease through its consequences. 

According to the validation studies, two groups of questions can be distinguished from the scale: 1–2 and 3–8 [44]. Questions 1–2 deal with individual assessments and abstract issues, while questions 3–8 confront patients with real-life problems. The AIS result may therefore be heterogeneous, so the total AIS score and responses in groups 1–2 and 3–8 were analyzed separately.

##### General Self-Efficacy Scale (GSES) 

GSES aims to assess adaptive potential challenging environmental demands by taking corrective action [45]. Because of the clear redundancy of all 10 questions, the scale was criticized as too homogenous and a short version of the GSES (GSE-6) was introduced with six items (2, 3, 5, 6, 7, 10, respectively) selected because of the highest coefficients of variation [46]. The GSE-6 was used in this study. 

#### 2.3.2. Other Measurements 

##### PANSS 

Clinical parameters were examined with the Positive and Negative Syndrome Scale (PANSS) [47]. This 30-item interview was conceived as an operationalized instrument that provides balanced representation of positive, negative, and general psychopathology in patients with schizophrenia. It consists of three subscales and a total score of psychotic severity.

##### Evoked Potentials 

The auditory evoked potentials (AEPs) were acquired using a Cognitrace neuro-psychiatry system. Twenty-four measurements consisted of the latencies to six alternating positive and negative peaks P50, N1, P2, N2, P3, and P4, and six amplitudes, respectively, in F-z and C-z locations. 

Twenty-one cup electrodes (10–20 international system) with ear and ground electrodes were used: Fp-z, F-z, C-z, P-z, O-z, Fp1, Fp2, F3, F4, C3, C4, P3, P4, O1, O2, F7, F8, T3, T4, T5, T6, A1, A2, and GND. Participants stayed in a separate, dark room. The test was performed with the subject in a sitting position, with eyes closed, and wearing earphones through which the acoustic stimuli were delivered in accordance with the oddball paradigm (a series of tones with frequencies in the range from 1000 Hz to 2000 Hz of 70 dB for 100 ms in a random sequence). One test lasted 3 min and 20 s and contained 80% of frequent stimuli and 20% of rare (target) stimuli. The subject was required to respond to the target stimuli by pressing the button. 

##### QEEG 

A Quantitative Electroencephalography–Neurofeedback (QEEG) was performed to map and meta-analyze recordings. The QEEG involved measuring a number of frequency bands and indices in different locations (34 measurements in total): delta (0.5–4 Hz), theta (4–8 Hz), alpha (8–12 Hz), beta (>12 Hz), SMR (sensorimotor rhythm, 12–15 Hz), beta1 (15–18 Hz), beta2 (18 Hz), gamma (40 Hz and above), theta/beta-attention factor, theta/SMR-concentration factor, SMR/beta2-tension and stress factor, alpha/SMR- sensory and motor activity factor, alpha/beta-executive function index, and beta/alpha-thinking and action factor. 

QEEG was performed twice, at the beginning of the experiment (T1) and after 3 months (T2) using the EEG Digi-Track ELMIKO apparatus (Elmico-Medical Company, Warsaw, Poland). The patients were tested with two electrodes, in the F-z and C-z regions and the Fast Fourier Transform (FFT) algorithm switched the raw EEG recording into QEEG power spectrum. 

### 2.4. Rehabilitation Therapy 

Our programme consisted of five main modules: social trainings, motivation/planning capacity, cognitive trainings, computer-assisted trainings (perception, attention, reasoning), and creativity module. It emphasizes not only teaching skills, but also improving metacognition and solving social problems. It was a largely balanced, psychosocial therapy programme, and the achievement of any particular skill was not an including or excluding criterion. Our rehabilitation program referred to some extent to the cognitive remediation therapy principles developed by Wykes et al., showing a predictive potential relating to the patient’s ability to function in the community [48].

The primary aim of the intervention was to improve social competence of the patients. The programme was administered to groups and was not hierarchically or sequentially organized. It was aimed at changing the daily routine by means of additional social activities, building team competences, training social roles, increasing personal acceptance, and strengthening one’s independence. Structured activities were held for up to 8-h blocks daily (except at weekends). The general plan of the day included group activities such as assertive training and role-playing techniques, psychotherapy, psychoeducation, cognitive training, art therapy, physiotherapy, sports, social events, cooking meals together, entertainment activities, and relaxation training. At least one session of group psychotherapy or psychoeducation was held every day.

### 2.5. Laboratory 

The serum level of NPS was determined immunoenzymatically with the ELISA technique (Human NPS/Neuropeptide S ELISA Kit, EIAab Science Co., 6618 h catalog number, Biopark, Opties Valley, Wuhan, China). The NPS level was determined at 07:00 a.m. (pg/mL), using a non-contact method of blood sampling into a clot tube. 

### 2.6. Statistical Analyses 

The values of the investigated variables were presented as means and standard deviations. The sociological and demographic parameters were presented as numbers and percentages. The results were compared using Student’s *t*-test for dependent samples, non-parametric Mann–Whitney U-test, Kruskal–Wallis *H*-test and Chi-squared test, as well as Pearson’s r product-moment correlation coefficient. The Shapiro–Wilk test was used to check whether samples came from a normal distribution. Differences were considered statistically significant at *p* < 0.05. Analyses were performed using Statistica 13.3. 

### 2.7. Ethical Issues 

The study protocol was approved by the local Bioethics Committee-approval no. KE-0254/35/2016. All the patients invited to take part in the study gave their written informed consent. 

## 3. Results 

The baseline (T1) and 3-month (T2) neurocognitive results of rehabilitation therapy (REH) versus standard therapy (CON) programs were presented in Table 2.

Of the results of five neurocognitive tests, only two of the tools-d2 and BCIS-had significant changes in time between T1 and T2. 5 out of 6 d2 indices in the REH group improved significantly (TN, E, E%, TN-E, CP). However, in the case of the CON group, improvement was noted only in the TN-E index. The BCIS-REF and BCIS-INDEX scores improved in the REH group, but there were no significant changes in the CON group. 

The results of the other three neurocognitive tests, CTT, AIS and GSES, did not change significantly over the three-month period neither in the REH group nor in the CON group. Also, the use of special, more measurement-specific variants of two tests (AIS 1–2/AIS 3–8 and GSES-6) did not change that.

In the REH group, the NPS serum level decreased significantly, in contrast to the CON group. The PANSS results turned out to be significantly different only for the Positive subscale in the REH group.

No significant differences were found in the REH and CON groups between T1 and T2 in terms of auditory evoked potentials (24 parameters in total). Similarly, there were no significant changes in the REH group in terms of QEEG (34 parameters in total). In the CON group, there were some sporadic differences in QEEG between T1 and T2 in indices mainly consisting of theta waveform (Fz theta/alpha, Fz theta/SMR, Fz alpha/theta). 

A comparison of the main effects of rehabilitation therapy over a period of 3 months is presented in Table 3.

Only some neurocognitive results differentiated REH and CON. What is a common rule for those indices is that they were the same scales (d2, BCIS) that showed any cognitive improvement over the 3-month study period.

In the REH group, the decrease in serum NPS levels was greater than in the CON group. All PANSS scores (Total, Positive, Negative, General) improved more in REH group than CON group. 

Supplementing the results of Table 3 with electrophysiological data, only one difference between REH and CON should be noted among all AEPs (out of 24 measurements) and QEEGs (out of 34 measurements):
QEEG/F-z: Theta/SMR index-practically no theta shares vs. SMR in REH group and significant theta share in CON (respectively: 0.04, SD 0.54 vs. 0.49, SD 0.54; *p* = 0.022),AEP P2/C-z/amplitude: the P2 waveform was reduced in REH and increased in CON after 3-month period (respectively: −2.83, SD 5.71 vs. 1.55, SD 3.85; *p* = 0.016).

All of REH group results which changed significantly during 3-month trial (Table 3, T2-T1 differences) were correlated with the NPS serum scores (Table 4). 

Analyzing socio-epidemiological parameters, two of them correlated strongly and significantly with the NPS reduction (T2-T1):
the increase in the number of **education** grades (r −0.67);the shorter duration of untreated psychosis (**DUP**) preceding the onset of schizophrenia (r −0.55).

## 4. Discussion 

According to Andreasen et al., schizophrenia should be understood directly as a neurocognitive disorder [1]. Nevertheless, while obvious cognitive impairments have been repeatedly demonstrated in patients with schizophrenia, not all the necessary elements of a complete pathophysiological theory have been established yet [4]. There is no unequivocal neural and biochemical basis. Attempts to directly treat cognitive dysfunctions in schizophrenia have not brought satisfactory results [5]. There is currently no consensus on the internal systematics of cognitive disorders in schizophrenia, which deficits are primary or secondary, how to separate simple “data metabolism” from sophisticated metacognition, how to calculate emotional and personality influences on virtually all aspects of cognition, what is the hierarchy of dysfunction, etc. [49]. Most of neurocognitive hypotheses, for lack of better ones, refer directly or indirectly to the half-century model of Baddeley and Hitch’s working memory [50]. In turn, some comprehensive models accept this “fuzzy logic” of reasoning and gather cognitive functions into several intentional processes, operating computationally and creating functional hierarchy [51]. In this context, research on cognitive dysfunction in schizophrenia still resembles classical 19th-century trial-and-error experiments. The results of our work provide some hints for those issues. Thus far, there have been no clinical studies with the primary goal of assessing the NPS serum level in relation to schizophrenia neurocognitive dysfunctions. The use of the structured rehabilitation therapy improved some, but not all, neurocognitive functions in schizophrenia. At the same time, a significant reduction in the serum NPS level was identified. 

### 4.1. NPS and Neurocognitive Results 

Only some neurocognitive tests responded positively to the rehabilitation therapy, what was at the same time related to significant NPS serum level reduction. Nearly all (five out of six) subtests of d2 were sensitive to 3-month therapy effects in REH group (except for FR; Table 2). Contrary to that, only one single response was noted in CON group. When compared head-to-head REH vs. CON results (Table 3), four out of six subtests of d2 in REH group specifically improved over CON group. Finally, the subtest valid mostly for tracking cognitive improvement and the relationship with NPS level was the TN-E complex score.

The TN-E is not just a “number of true responses”, but rather the index of two balanced and integrated mental activities The first is a time-dependent, goal-oriented, and learning-while-performing activity. Consequently, it is similar to CTT or Trail Making Test (TMT)-like measurements [52,53]. The second is the potential to correct oneself, i.e., to avoid both omission and commission errors. As a result, the ability to manage contradictory patterns heavy weights on TN-E results. The final score does not depend on speed process alone but is the balanced measurement of final accuracy or problem solving in general. The TN-E is therefore not a measurement of trained activity, but it verifies the functioning strictly in metacognitive capacity [54,55]. It is important as metacognitive functions have not been tested sufficiently in schizophrenia so far, because of their measurement complexity [51,56]. 

Similarly, using two BCIS indices-Self-Reflectiveness and Reflectiveness-Certainty Index-a strong correlation of pro-cognitive effects with the NPS level was confirmed (Table 3). Like TN-E, the BCIS Index is a composite, internally confronted measure. This index verifies the ability to detect and correct misinterpretations diagnosed in patients using the Self-Certainty subscale. This time again, it is about the assessment of a complex mental process in which the final behavioral optimization is the result of a balance between the pursuit of assertive action in confrontation with the need to avoid psychotic experiences and anomalous beliefs. The Reflectiveness–Certainty Index structure is therefore appropriate for measuring metacognition.

The characteristics of the Self-Reflectiveness index from the BCIS are slightly different. It was also significantly and strongly correlated with the NPS level. Self-Reflectiveness is a collection of assessments that verify what the patient thinks other people think and feel about the patient’s personal behavior. Therefore, it is a direct measurement of the ability to be empathetic, and more broadly, a measurement of the patient’s social cognition resources [57]. While the *G12-lack of judgment and insight* from PANSS examines the insight only into the diagnosis and treatment itself, the BCIS provides a broader range of cognitive interpretations, including understanding of the patient’s own situation and attitudes from the social environment, the ability to distinguish true-or-false experiences, and the level of assertiveness [58]. 

The essence of the positive correlation of the above-listed neurocognitive results and NPS would be a significant relationship that appear only using measurement tests specific for phenomena such as metacognition and social cognition. This also explains the “inactivity” of the remaining tests:
-CTT is a tool that verifies the ability to focus and maintain attention while performing one short task. This scheme is not changed by the slightly more task-related variant of the CTT-2. CTT does not require taking a position on any social context and is basically a dexterity test like many computer games. The CTT-2 variant in relation to the CTT-1 requires only a slightly greater “inhibition” of the competing instructions because there are only two instructions in total. This level of “inhibition” does not require complicated metacognition schemas to be used. Nevertheless, CTT or TMT tests are very sensitive in identifying cognitive disorders in schizophrenia [52,53], but these are not tools for verifying cognitive strategy building as in the case of problem-solving tests (e.g., Wisconsin Card Sorting Test, Tower of London);-AIS is a tool for the comprehensive assessment patient’s attitude to his/her disease, but the structure of the scale, including its excessive redundancy is monothematic and actually includes what in fact a single *G12-lack of judgment and insight* from PANSS can offer [47]. The issue of the patient’s lack of insight into suffering from schizophrenia is so fundamental that it defines this disease. Thus, a paradoxical methodological problem is that a patient who consistently disagrees with the diagnosis of schizophrenia would be instructed to freely reflect doubts on the AIS list of statements. Actually, it would be resolved at the level of defense mechanisms, and not of any cognitive flexibility processes. Of course, the AIS scale may be successfully used in the self-assessment of patients with psychosomatic diseases, where this type of paradox does not occur. Finally, it should be added that using the suggested methodological split into two groups of questions (1–2 and 3–8), it was also not possible to modify the results [44]; -GSES can determine in patients, especially those with psychosis, a “defensive” type of response resulting from a sense of their own disability. A patient with psychosis does not have sufficient cognitive capacity to relativize his/her psychotic position. Therefore, this would be a methodologically similar problem to that discussed in the case of applying AIS. In this situation, statements that sound almost identical and are repeated ten times, can only reinforce a defensive attitude. The scale was criticized because of this as being too homogenous and a short version of the GSES (GSE-6) was proposed with six items [46], however even by applying this modification we could not change the overall scale specificity that does not reach the metacognition spectrum.

The study managed to show a specific association of NPS with metacognition and social cognition tests. However, other neurocognitive tests were not effective when the challenge was to measure complex cognitive behavior (CTT for perceptual tracking and sequencing only, AIS and GSES operating below the threshold of defense mechanisms). Paradoxically, the cognitive impairment in schizophrenia is so generalized that using tests related to practically every cognitive domain we can differentiate the results of healthy and sick people [4,52,53]. However, due to this “excessive test sensitivity”, it is problematic to carry out any specific measurement in schizophrenia, and more generally, to settle the model of integration of cognitive functions in schizophrenia [51]. That is why it was so important to establish in this study a significant relationship between the three neurocognition indices and the NPS level. 

Finally, it should be emphasized that some socio-epidemiological parameters (education, duration of untreated psychosis) that accompanied the reduction of NPS and are commonly understood as co-factors of cognitive functioning in schizophrenia [1,4]. 

### 4.2. Clinical Results 

A structured, 3-month rehabilitation therapy programme was implemented in the REH group, with partial improvement in clinical outcomes (Table 2: PANSS General, PANSS Total). The clinical results were even more favorable in the direct comparison of the REH and CON groups (Table 3). A significant correlation of NPS T1 and NPS T2–T1 was affiliated only to General and Total PANSS results (Table 4), but not to its more specific positive and negative factors. However, since the General subscale is a collection of a variety of symptoms, the result could not be tracked further based on the original PANSS model. For further analysis of PANSS symptomatology versus NPS serum level see Markiewicz-Gospodarek et al. (2022) [8]. 

The clinical effectiveness of the rehabilitation programme may have been due to the fact that it was not solely focused on cognitive training, but was a more complex, long-term psychosocial therapy. The meta-analysis of rehabilitation techniques shows that they have an impact on global functioning and improvement in psychopathology only if they implement integrated psychosocial rehabilitation [49]. Improvement of cognitive functions, even if it occurs, may be not a sufficient condition to obtain positive clinical effects.

### 4.3. Electrophysiology 

It is assumed that the use of modern methods of electrophysiological diagnostics would benefit from biomarkers that will provide sensitive and reliable measurements of the neural events underlying cognitive dysfunctions in schizophrenia. However, so far, no unequivocal results have been obtained in this respect [59]. In our work, it was not possible to link cognitive variables and changes in the NPS level with the results of several dozen measurements using two basic methods of modern electrophysiological diagnostics (AEP, QEEG). This may be due to the structure of the study itself (long-term, 3-month), as well as the fact that the improvement of cognitive functions related specifically to metacognition or social cognition variables, and not to simple cognitive deficits.

### 4.4. Study Limitations

Research on cognitive functions in schizophrenia has been going on for over a century and is associated with a variety of concepts, tools, and limitations [2]. The neurocognitive approach assumes the connection of cognitive phenomena with neurophysiological substrates [1]. In works of this type, even such complex phenomena as insight, metacognition or social cognition are being examined [60,61]. This approach is of a research nature, and thus the results have their limitations, and entail changes in the methodology. This also applies to our work.

Relatively often used in schizophrenia research cognitive batteries such as MATRICS and BACS were not administered in our study as this would not be consistent with the main goals due to methodological limitations. We were focused on patients with a specific and dynamic clinical profile, while the results of MATRICS turned out to be only minimally related to clinical symptom type and schizophrenia severity [62], and in turn, BACS measurement has not been validated in relation to the longitudinal relationship of cognition with functional capacity, real-world functional outcome, and schizophrenia treatment [63].

The presented study confirmed the serum NPS level as a phenomenon accompanying the improvement of certain cognitive functions during treatment of patients with schizophrenia. This relationship, based on patients’ clinical improvement, enables better treatment planning and prognosis. However, the study had some clear limitations: small groups, only men recruited, only the subtype episodic schizophrenia with stable or progressive development of negative symptoms and focus on rehabilitation effects. This does not allow us to draw unambiguous conclusions and means that the verification of all conclusions requires the extension of the study. Nevertheless, the results are pioneering the possible association of NPS (neuropeptides) with cognitive functions in schizophrenia and should be carefully considered as a chance to meet the diagnostic and therapeutic needs of patients. 

## 5. Conclusions 

(1)By using the long-term structured rehabilitation therapy in patients with schizophrenia, an improvement in selected cognitive functions was achieved, accompanied by a decrease in the level of neuropeptide S (NPS) in the serum;(2)The primary effect was specific to the cognitive improvement described by specific test results:
TN-E—combined score of the total responses minus omission and commission errors of d2 Sustained-Attention Test;Self-Reflectiveness score and Reflectiveness-Certainty Index of Beck Cognitive Insight Scale.
(3)Reduction of NPS, a neuropeptide associated with clinical disorganization in schizophrenia, has been associated with improved cognitive functioning in domains of metacognition and social cognition after 3-month rehabilitation therapy;(4)The primary effect was related to the current improvement in the clinical condition (PANSS) and the course of schizophrenia (education, duration of untreated psychosis);(5)The cognitive effects depending on the NPS level could not be associated with the results of QEEG and AEP measurements.

## Figures and Tables

**Table 1 jcm-11-05266-t001:** Initial (T1) parameters and pairwise comparisons (*t* test/Mann–Whitney test) for REH, CON and Non-clinical groups.

(**A**)												
**Variable**	**REH**			**CON**		**REH vs. CON**		**Non-Clinical**			**REH vs. Non-Clinical**	
	**M**	**SD**		**M**	**SD**	**t ^t^/U ^U^**	** *p* **	**M**		**SD**	**t ^t^/U ^U^**	** *p* **
d2-TN	304.63	36.99		330.65	36.99	1.79 ^t^	0.083					
d2-Errors	144.06	23.51		137.71	23.51	−0.42 ^t^	0.675					
d2-%Errors	47.71	8.71		43.23	8.71	−0.79 ^t^	0.436					
d2-TN-E	160.56	38.83		192.94	38.83	1.45 ^t^	0.158					
d-CP	110.69	29.77		134.06	41.17	130.00 ^U^	0.843					
d-FR	15.25	29.77		15.94	41.17	87.00 ^U^	0.081					
CTT-1	60.56	24.74		58.94	26.03	127.50 ^U^	0.773					
CTT-2	126.06	39.58		123.12	55.48	120.50 ^U^	0.589					
CTT-II	1.19	0.59		1.12	0.51	126.00 ^U^	0.732					
BCIS-REF	20.81	3.53		22.94	5.26	114.00 ^U^	0.439					
BCIS-CER	14.44	0.99		16.12	3.77	1.28 ^t^	0.210					
BCIS-INDEX	6.38	2.50		6.82	4.07	0.38 ^t^	0.707					
AIS-Total	26.44	9.12		29.06	6.98	0.93 ^t^	0.359					
AIS (1–2 items)	6.69	2.47		7.00	2.53	121.50 ^U^	0.614					
AIS (3–8 items)	19.75	7.25		22.06	5.15	1.06 ^t^	0.298					
GSES-Total	28.88	7.25		31.59	6.27	1.15 ^t^	0.258					
GSES-6 (items)	17.81	4.21		19.00	3.89	119.50 ^U^	0.564					
PANSS Total	53.13	7.29		53.41	15.73	119.0 ^U^	0.552					
Age of first hospitalization (years)	22.69	3.36		25.12	5.10	1.61 ^t^	0.119					
DUP (years)	2.00	1.10		2.59	1.37	1.35 ^t^	0.185					
Education (ISCED grades)	3.50	1.10		3.35	0.49	−0.05 ^t^	0.619					
Antipsychotics in milligrams (equivalents of olanzapine)	21.28	6.88		19.32	4.97	121.5 ^U^	0.614					
NPS (pg/mL)	48.46	16.32		39.67	7.14	82.5 ^U^	0.061	42.97		16.55	64.0 ^U^	0.360
BMI (kg/m^2)^	29.84	4.05		27.39	2.81	−2.02 ^t^	0.052	28.85		3.88	0.69 ^t^	0.496
Age (years)	36.00	7.79		39.35	10.65	1.03 ^t^	0.312	41.27		7.48	−1.92 ^t^	0.065
(**B**)												
**Variable**	**REH**				**CON**		**Non-Clinical**				***H*-Test**	
	**Ranks Sum**		**Ranks Mean**		**Ranks Sum**	**Ranks Mean**	**Ranks Sum**		**Ranks Mean**		** *H* **	** *p* **
NPS (pg/mL)	479.5		29.97		400.5	23.56	296.0		19.73		4.26	0.1189
BMI (kg/m^2^)	464.0		29.00		345.0	20.29	367.0		24.47		3.19	0.2032
Age (years)	310.0		19.38		437.5	25.74	428.5		28.57		3.56	0.1690

(**A**): REH—patient rehabilitation group, CON—patient control group, Non-clinical—healthy reference group, d2-TN—d2 total number, d2-E—d2 errors, d2-E%—d2 percentage of all errors, d2-TN-E—d2 total number minus all errors, d2-CP—concentration performance, d2-FR—d2 fluctuation rate, CTT—Color Trails Test, CTT-II—interference index, BCIS—Beck Cognitive Insight Scale (reflectiveness, certainty), AIS—Acceptance of Illness Scale, GSES—General Self-Efficacy Scale, PANSS Total—total result of Positive and Negative Syndrome Scale, DUP—duration of untreated psychosis, ISCED—International Standard Classification of Education, NPS—Neuropeptide S, BMI—body mass index, M—mean, SD—standard deviation, ^t^—Student’s *t*-test, ^U^—Mann-Whitney U-test, *p*—*p*-value significance at *p* < 0.05. (**B**): REH—patient rehabilitation group, CON—patient control group, non-clinical—healthy reference group, NPS—Neuropeptide S, BMI—body mass index, *H*-test—Kruskal-Wallis *H*-test by ranks, *p*—*p*-value significance at *p* < 0.05.

**Table 2 jcm-11-05266-t002:** T1 versus T2 neurocognitive tests, NPS and PANSS results.

Test	Subtest	Group	Baseline		Final		t/U	*p*
			M	SD	M	SD		
d2	TN	REH	304.63	36.99	361.31	36.02	−4.39 ^t^	**0.000**
		CON	330.65	45.63	365.53	63.59	−1.84 ^t^	0.075
	Errors	REH	144.06	23.51	77.75	18.47	8.87 ^t^	**0.000**
		CON	137.71	55.53	107.53	33.31	1.92 ^t^	0.064
	% Errors	REH	47.71	8.71	21.64	5.11	10.33 ^t^	**0.000**
		CON	43.23	21.03	30.72	12.55	94.00 ^U^	0.085
	TN-E	REH	160.56	38.83	283.56	38.26	−9.03 ^t^	**0.000**
		CON	192.94	81.13	258.00	74.72	−2.43 ^t^	**0.021**
	CP	REH	110.69	29.77	153.75	53.04	−2.83 ^t^	**0.008**
		CON	134.06	41.17	134.82	40.10	−0.06 ^t^	0.957
	FR	REH	15.25	9.66	14.00	6.81	126.00 ^U^	0.955
		CON	15.94	9.43	15.65	10.74	135.00 ^U^	0.757
CTT	CTT-1	REH	60.56	24.74	55.56	19.63	110.50 ^U^	0.522
		CON	58.94	26.03	56.08	20.02	141.50 ^U^	0.931
	CTT-2	REH	126.06	39.58	114.81	33.73	101.50 ^U^	0.327
		CON	123.12	55.48	113.28	45.48	136.50 ^U^	0.796
	CTT-II	REH	1.19	0.59	1.14	0.43	124.00 ^U^	0.895
		CON	1.12	0.51	1.10	0.57	0.10 ^t^	0.918
BCIS	BCIS-REF	REH	20.81	3.53	24.44	3.76	−2.81 ^t^	**0.009**
		CON	22.94	5.26	20.18	5.15	1.55 ^t^	0.131
	BCIS-CER	REH	14.44	3.76	14.69	3.79	−0.19 ^t^	0.853
		CON	16.12	3.77	16.53	3.22	−0.34 ^t^	0.735
	BCIS-INDEX	REH	6.38	2.50	9.75	3.26	51.00 ^U^	**0.004**
		CON	6.82	4.07	3.65	6.22	1.76 ^t^	0.088
AIS	Total	REH	26.44	9.12	26.63	7.98	−0.06 ^t^	0.951
		CON	29.06	6.98	30.59	7.04	−0.64 ^t^	0.529
	AIS (1–2)	REH	6.69	2.47	6.31	2.91	120.50 ^U^	0.792
		CON	7.00	2.53	7.47	2.07	133.50 ^U^	0.718
	AIS (3–8)	REH	19.75	7.25	20.31	6.65	−0.23 ^t^	0.821
		CON	22.06	5.15	23.12	5.57	−0.58 ^t^	0.569
GSES	Total	REH	28.88	7.25	31.69	4.98	−1.28 ^t^	0.211
		CON	31.59	6.27	32.00	6.38	−0.19 ^t^	0.851
	GSES-6	REH	17.81	4.22	19.25	2.89	117.50 ^U^	0.706
		CON	19.00	3.89	19.06	4.01	−0.04 ^t^	0.966
NPS (pg/mL)		REH	48.46	16.32	36.01	3.45	34.00 ^U^	**0.000**
		CON	39.67	7.14	38.96	6.76	134.00 ^U^	0.731
PANSS	Total	REH	53.13	7.29	48.50	8.22	−1.68	0.103
		CON	53.41	15.73	57.88	7.40	1.06	0.297
	Positive	REH	9.75	1.73	8.25	1.39	−2.70	**0.011**
		CON	10.00	2.40	9.88	8.25	−0.12	0.906
	Negative	REH	15.44	3.46	14.00	3.39	−1.19	0.245
		CON	15.29	3.64	16.65	2.71	1.23	0.228
	General	REH	27.94	3.55	26.25	4.51	−1.18	0.2445
		CON	28.12	10.83	31.35	3.20	1.18	0.246

d2—d2 Sustained-Attention Test, TN—total number of letters marked, E—errors, E%—percentage of all errors, TN-E—total number of items processed minus errors, CP—concentration performance, FR—fluctuation rate, CTT—Color Trails Test, CTT-II—Interference Index, BCIS—Beck Cognitive Insight Scale, BCIS-REF—self-reflectiveness subscale, BCIS-CER—self-certainty subscale, AIS—Acceptance of Illness Scale, GSES—General Self-Efficacy Scale, NPS—neuropeptide S, PANSS—Positive and Negative Syndrome Scale, ^t^—Student’s *t*-test, ^U^—Mann-Whitney U-test.

**Table 3 jcm-11-05266-t003:** Differences in the magnitude of change from pre- (T1) to post-therapy (T2) results in REH and CON groups.

Test	Subtest	REH (T2-T1)		CON (T2-T1)		In-between Comparisons	
		M	SD	M	SD	t/U	*p*
d2	TN	56.69	31.73	34.88	41.09	−1.70 ^t^	0.100
	Errors	−66.31	24.22	−30.18	39.52	3.14 ^t^	**0.004**
	% Errors	−26.08	10.09	−12.51	15.25	3.00 ^t^	**0.005**
	TN-E	123.00	47.52	65.06	66.85	−2.85 ^t^	**0.008**
	CP	43.06	43.72	0.77	43.02	−2.80 ^t^	**0.009**
	FR	43.06	5.98	−0.29	6.98	0.42 ^t^	0.678
CTT	CTT-1	−5.00	14.94	−2.86	13.71	0.42 ^t^	0.671
	CTT-2	−11.25	27.81	−9.85	34.04	0.16 ^t^	0.875
	CTT-II	−0.05	0.70	−0.02	0.73	0.14 ^t^	0.893
BCIS	BCIS-REF	3.63	5.10	−2.77	6.82	−3.03 ^t^	**0.005**
	BCIS-CER	0.25	4.77	0.41	3.81	125.00 ^U^	0.719
	BCIS-INDEX	3.38	3.59	−3.18	7.15	56.00 ^U^	**0.004**
AIS	Total	0.19	9.54	1.53	5.64	0.50 ^t^	0.624
	AIS (1–2)	−0.38	3.34	0.47	2.32	0.85 ^t^	0.403
	AIS (3–8)	0.56	9.04	1.06	4.44	0.20 ^t^	0.841
GSES	Total	2.81	8.34	0.41	4.54	118.00 ^U^	0.528
	GSES-6	1.44	5.05	0.06	1.44	116.50 ^U^	0.494
NPS (pg/mL)		−12.46	15.97	−0.72	9.97	71.00 ^U^	**0.020**
PANSS	Total	−4.63	3.40	4.47	10.93	23,50 ^U^	**0.000**
	Positive	−1.50	1.26	−0.12	1.32	59,50 ^U^	**0.006**
	Negative	−1.44	1.46	1.35	3.06	3.31 ^t^	**0.002**
	General	−1.69	2.02	3.24	9.08	23,00 ^U^	**0.000**

d2—d2 Sustained-Attention Test, TN—total number of letters marked, E—errors, E%—percentage of all errors, TN-E—total number of items processed minus errors, CP—concentration performance, FR –fluctuation rate, CTT—Color Trails Test, CTT-II—Interference Index, BCIS—Beck Cognitive Insight Scale, BCIS-REF—self-reflectiveness subscale, BCIS-CER—self-certainty subscale, AIS—Acceptance of Illness Scale, GSES—General Self-Efficacy Scale, NPS—neuropeptide S, PANSS—Positive and Negative Syndrome Scale, ^t^—Student’s *t*-test, ^U^—Mann-Whitney U-test.

**Table 4 jcm-11-05266-t004:** The Pearson’s r product–moment correlation coefficients: NPS T1, NPS T2, and NPS T2-T1 correlated with neurocognitive and physiological variables (T2-T1). Only strong correlations for absolute values of r > 0.5 (*p* < 0.05) were bolded. *p*-values in parentheses.

Variable (T2-T1 Difference)		NPS T1	NPS T2	NPS T2-T1
d2	Errors	−0.01 (0.469)	0.17 (0.787)	0.05 (0.424)
	%Errors	0.03 (0.650)	0.40 (0.322)	0.07 (0.802)
	TN-E	−0.21 (0.279)	**−0.68 (0.007)**	0.05 (0.563)
	CP	−0.35 (0.244)	−0.32 (0.043)	0.30 (0.445)
BCIS	BCIS-REF	−0.55 (0.117)	0.12 (0.738)	**0.62 (0.019)**
	BCIS-INDEX	0.01 (0.249)	0.38 (0.212)	0.09 (0.142)
PANSS	Total	0.45 (0.215)	−0.33 (0.383)	**−0.54 (0.040)**
	Positive	0.32 (0.430)	−0.34 (0.376)	−0.40 (0.315)
	Negative	−0.03 (0.824)	−0.15 (0.649)	0.00 (0.897)
	General	**0.54 (0.017)**	−0.20 (0.564)	**−0.60 (0.045)**
QEEG/F-z	Theta/SMR index	−0.44 (0.085)	−0.47 (0.461)	0.35 (0.114)
AEP/C-z	P2 amplitude	0.42 (0.280)	−0.07 (0.712)	−0.47 (0.234)

d2—d2 Sustained-Attention Test, E—errors, E%—percentage of all errors, TN-E—total number of items processed minus errors, CP—concentration performance, BCIS—Beck Cognitive Insight Scale, BCIS-REF—self-reflectiveness subscale, NPS—neuropeptide S, PANSS—Positive and Negative Syndrome Scale, QEEG—Quantitative Electroencephalography, AEP—Auditory evoked potentials.

## Data Availability

Not applicable.

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
