# Peer review of "Relationship of Neuropeptide S (NPS) with Neurocognitive, Clinical, and Electrophysiological Parameters of Patients during Structured Rehabilitation Therapy for Schizophrenia"

_jcm, 2022, doi:10.3390/jcm11185266_

Round 1

Reviewer 1 Report

In general, the manuscript leaves a good impression. The research topic is relevant and has a high degree of novelty. A significant effect of structured rehabilitation therapy was shown, confirmed by a laboratory marker. But the main disadvantage of the work is very small clinical groups. This does not allow us to draw unambiguous conclusions. The authors justify the small size of the groups when describing the objects of study. But this is the main limitation of the work. Therefore, it should be placed in a separate paragraph "limitation" and described in detail. 

2) There is no complete clarity with statistical research methods. What methods were used papometric or non-parametric? The tables present the values of the mean and standard deviation, which means that it is assumed that the results obeyed the normal distribution law. But this is very doubtful for such small groups. I would like to see evidence.

 3) In the course of the study, the results of the three groups were compared. But only pairwise comparison results are presented. Show p values for Kruskal–Wallis test.

4) The values of NPS for the group of patients with REH after therapy are very close to the values of the CON group before the start of the study. Show their p values. Why are the NPS values in both groups before the start of the experiment significantly different?

5) Make the conclusion more detailed.

Author Response

Dear Reviewer,

Thank you very much for reviewing our manuscript. We appreciate the interest and commitment you have provided for this work. We are very grateful for your extremely precious comments. We are convinced that thanks to your suggestions this manuscript will be much more valuable.

We are pleased to submit explanations and details of our revisions in the manuscript entitled “Relationship of neuropeptide S (NPS) with neurocognitive, clinical, and electrophysiological parameters of patients during structured rehabilitation therapy for schizophrenia.

The followings are our point-by-point responses:

  1. We clearly marked which comparisons were made with parametric or non-parametric tests, marking the tests with the letter t or U respectively. For example, in Table-1, sixteen comparisons were made with Student's t-test and twelve with U-Mann-Whitney test. The choice of the test was determined each time by the parametric assessment carried out with the Shapiro-Wilk test. The procedure is clearly described in chapter 2.6. Statistical analyzes
  2. As requested, we additionally provided comparisons with the Kruskal-Wallis’s test, introducing the new Table 1A
  3. Certainly, NPS values were not significantly different before the start of the experiments - see both Table 1A and Table 1B
  4. Conclusions were discussed more detail, in particular in the new chapter Study limitations

We hope that after this revision, the manuscript is of a higher quality and worth reading.

We wish you all the best!

Sincerely,
Agnieszka Markiewicz – Gospodarek
on behalf of all authors

Reviewer 2 Report

In this study, the authors describe the beneficial effect of structured rehabilitation therapy on schizophrenic patients, both in neurocognitive tests and in clinical evaluations, and their relationship with neuropeptide S as a biomarker in schizophrenia. 

The paper is interesting and innovative, although the work may be improved by minor review in different points:

+INTRODUCTION:

-The authors make a very clear and deep summary of the knowledge about neuropeptide S and its receptor and their role in schizophrenia. However, it should be spelling checked to detect and change some expressions, for example: “the studies of this type” (line 67, page 2), etc.

+MATERIALS AND METHODS:

-The authors say that 16 and 17 patients were included in the intervention and control groups, respectively (line 141, page 3), but the abstract describes that each group consisted of 15 patients (lines 23-24, page 1). The authors should clearly explain these differences. Also, in line 142, page 3, they point that “a safe rule of thumb suggested a minimum sample size of 2x15 (REH, CON)”. Explain this more deeply.

-In lines 189-190, page 5 it is said that: “only some of them [patients] additionally received typicals [antipsychotics] (sulpiride, perazine, zuclopenthixol, fluanxol, haloperidol; respectively: CON - 11.8%, REH - 20.0%)”. The authors must explain and discuss the importance of this difference and the implications that it may have in the observed outcomes. Also, write “flupentixol” instead of “fluanxol” because it is a commercial name.

+RESULTS

-Table 2, last row: it is said “Genral” instead of “General”.

+CONCLUSIONS

-Line 562, page 14 (conclusion 3): Explain that in this case “long term reduction of NPS” is referred to 3 months patient following.

Author Response

Dear Reviewer,

Thank you very much for reviewing our manuscript. We appreciate the interest and commitment you have provided for this work. We are very grateful for your extremely precious comments. We are convinced that thanks to your suggestions this manuscript will be much more valuable.

We are pleased to submit explanations and details of our revisions in the manuscript entitled “Relationship of neuropeptide S (NPS) with neurocognitive, clinical, and electrophysiological parameters of patients during structured rehabilitation therapy for schizophrenia.

The followings are our point-by-point responses:

  1. We corrected all minor editorial errors. Thank you for such a meticulous analysis of the text.
  2. We re-edited the entire paragraph on the pharmacotherapy used, carrying out additional statistical analysis (chi2 test) to avoid the risk of missing the effect of specific drug groups. The differences turned out to be insignificant.

We hope that after this revision, the manuscript is of a higher quality and worth reading.

We wish you all the best!

Sincerely,

Agnieszka Markiewicz – Gospodarek

on behalf of all authors

Reviewer 3 Report

This paper provided interesting results regarding the role of neuropeptide S on the relation between changes in cognitive functioning and a rehabilitation program in patients living with Schizophrenia. 

A major concern regards the definition provided by the authors of Cognitive Remediation Therapy (CRT). Indeed, the authors stated that remediation program consisted of five main modules (social trainings, motivation/planning capacity, cognitive trainings, computer-assisted trainings, and creativity module). This definition is quite confusing the reader since the original Cognitive Remediation Therapy by Wykes and Reeders, 2005 consisted in three main modules to enhance cognitive flexibility, working memory and planning and is not primary intended to improve social competence, as described in the manuscript. Moreover CRT is an individualized, drill & pencil strategy and not a computer-based intervention. Also the duration of Rehabilitation Therapy is different from the original CRT duration.

See Wykes, T. and Reeder, C. (2005) Cognitive Remediation Therapy for Schizophrenia: Theory and Practice. Brunner-Routledge, London.

Moreover, a major limitation is to not having included a structured evaluation of cognitive functioning with wide instruments such as BACS or MATRICS Batteries. An explanation for these lacks is needed by the authors and also this issue needs to be considered as a main limitation of the present study.

Beck Cognitive Insight Scale (BCIS) is described as a neurocognitive test, but this is quite incorrect since cognitive insight does not represent a strictly evaluation of cognitive functions. The same concern is for the Acceptance of Illness Scale (AIS). Moreover when the author referred to neuro-cognitive it could be misinterpreted since the term “neuro” generally refers to the “cold” cognitive functioning, while the term “hot” is generally related to the social cognition domain. An overall revision along all the manuscript should be welcomed. 

Another major limitation is about the lack of an adequate a priori power calculation to define the number of subjects required per arm. The authors reported that “Since we planned to analyse one independent variable at a time, a safe rule of thumb suggested a minimum sample size of 2x15”.

In the Results, to increase the readability f the results, I suggest to add the p-value for each clinical and cognitive variable changed during the study also in the text of the manuscript.

In the Results, Table 4 is quite confusing. I suggest editing this table reporting both values of r and p from the Pearson’s product-moment correlation analyses. Moreover, the result of BCIS - Self Certainty is lacking. 

Moreover, at page 10, lines 384-385 the results of these analyses should be revised in a more systematic manner. 

Discussion section:

Lines 421-22. this sentence is generic and should be rephrased according to the above suggestions. 

Line 550. The authors observed that “The study managed to show a specific association of NPS with metacognition and social cognition tests”. This is a strong inference and should be carefully reconsidered by considering that the tasks here applied are not specifically designed to detect metacognition and social cognition changes. Moreover, a strong limitation is to not having included a structured evaluation of these domain impairments with dedicated batteries and assessment tools. 

Similarly, the conclusions provided in the abstract (lines 36-38) should be softened.

The paper lacks of a limitation section that should be implemented.

Author Response

Dear Reviewer,

Thank you very much for reviewing our manuscript. We appreciate the interest and commitment you have provided for this work. We are very grateful for your extremely precious comments. We are convinced that thanks to your suggestions this manuscript will be much more valuable.

We are pleased to send a revised version of the manuscript entitled “Relationship of neuropeptide S (NPS) with neurocognitive, clinical, and electrophysiological parameters of patients during structured rehabilitation therapy for schizophrenia.

The followings are our point-by-point responses:

  1. At work, we implemented our original rehabilitation therapy program. The reference to the historical achievements of Cognitive Remediation Therapy was to - as clearly indicated - refer only to ideological principles, not the schemes themselves or the strict implementation of the program. The text has been corrected so that there is no doubt in this regard.
  2. In order to clarify the methodological doubts, the chapter "Study limitations" was introduced and discussed the issue of the different understanding of neurocognitive research on schizophrenia and the resulting diversification of methodological approaches. Research on cognitive functions in schizophrenia has been going on for over a century and is associated with a variety of concepts, tools, and limitations. The issue of the naming and classification of (neuro) cognitive disorders in schizophrenia remains the subject of changes and scientific discussion. The neurocognitive approach assumes the connection of cognitive phenomena with neurophysiological substrates. In works of this type, even such complex phenomena as in-sight, metacognition or social cognition are being examined. This approach is of a research nature, and thus the results have their limitations, and entail changes in the methodology. This also applies to our work. 
  3. Relatively often used in schizophrenia research cognitive batteries such as MATRICS and BACS were not administered in our study as this would not be consistent with the main goals due to methodological limitations. We were focused on patients with a specific and dynamic clinical profile, while the results of MATRICS turned out to be only minimally related to clinical symptom type and schizophrenia severity, and in turn, BACS measurement has not been validated in relation to the longitudinal relationship of cognition with functional capacity, real-world functional outcome, and schizophrenia treatment.
  4. P-values are clearly presented for each clinical and cognitive variable
  5. P-values were added in Table 4 according to the Reviewer
  6. The result of BCIS - Self Certainty is lacking in Table 4 as only results that changed significantly during 3-month trial (Table 3, T2-T1 differences) were presented. That was clearly explained in the text.

We hope that after this revision, the manuscript is of a higher quality and worth reading.

We wish you all the best!

Sincerely,

Agnieszka Markiewicz – Gospodarek

on behalf of all authors

Reviewer 4 Report

This article is a very important step in understanding the neurocognitive associations of Schizophrenia. The introduction part is very good with the article starting by describing the cognitive model of SCZ and the abnormalities of the neuropeptide system in SCZ.

The results interpretation and conclusion needs some work. The study shows that the group which was exposed to the rehabilitation therapy showed improved indices in d2 and BCIS and the PANSS scores and decrease in NPS serum levels.  To conclude that decreased serum levels of NPS was associated with improvement in metacognition and social cognition domains as well as clinical improvement in SCZ patients is quite a leap. I am not sure how this conundrum can be addressed.

Author Response

Dear Reviewer,

Thank you very much for reviewing our manuscript. We appreciate the interest and commitment you have provided for this work. We are very grateful for your extremely precious comments. We are convinced that thanks to your suggestions this manuscript will be much more valuable.

We are pleased to send a revised version of the manuscript entitled “Relationship of neuropeptide S (NPS) with neurocognitive, clinical, and electrophysiological parameters of patients during structured rehabilitation therapy for schizophrenia.

We hope that after this revision, the manuscript is of a higher quality and worth reading.

We wish you all the best!

Sincerely,

Agnieszka Markiewicz – Gospodarek

on behalf of all authors

Round 2

Reviewer 3 Report

The authors responded adequately to all requests. 

I suggest the acceptance of the present manuscript.